# The impact of wartime conflict on the mental health problems of women in the conflict-hit population in Woldia, Ethiopia

**Negussie Deyessa** [1]*, **Kehali Getahun**[2], **Ruth Filimona**[3], **Abraraw Tadesse**[4]

1 Department of Epidemiology and Biostatistics, School of Public Health, College of Medicine and Health Sciences, University of Rwanda, Kigali, Ethiopia, 2 School of Medicine, College of Health Sciences, Addis Ababa University, Addis Ababa, Ethiopia, 3 Department of Emergency, Gynecology and Obstetrics, Gandhi Memorial Hospital, Addis Ababa, Ethiopia, 4 Amhara Regional Health Bureau, Bahirdar, Ethiopia

* negdaysun@gmail.com

## Abstract

### Background

Common mental health problems are of significant public health importance, with severe social and economic impacts that adversely affect individuals' quality of life. The burden of these problems may worsen during wartime. This study aims to assess the prevalence of war-related common mental health problems, including depression, anxiety disorder, phobia, and posttraumatic stress disorder (PTSD), among women in the Woldia district, Amhara, Ethiopia.

### Methods

A community-based cross-sectional survey was conducted from February to March 2023, involving 1,505 eligible women selected from five kebeles using cluster followed by systematic sampling. The study used the Patient Health Questionnaire-9, Generalized Anxiety Disorder scale, and PTSD Checklist-5 to assess common mental health disorders. In addition to descriptive analysis, the study employed binary and multivariable analysis to evaluate sociodemographic correlates and the presence of comorbidity for each mental disorder.

### Result

Almost half of the women exhibited symptoms of common mental health problems, with about 33% experiencing comorbidity of two or more disorders. Depressive symptoms were more prevalent among older, single women and those with spouses using khat, whereas higher wealth and strong social support were protective factors. Similar risk and protective patterns were observed for generalized anxiety disorder (GAD) and posttraumatic stress disorder (PTSD). Older age, single status, and

**Data availability statement:** All relevant data are within the paper. Or you can find in the following repository Deyessa, Negussie (2025), "Woldia war time violence against women", Mendeley Data, V1, doi: 10.17632/w9gzzr4txj.1.

**Funding:** The author(s) received no specific funding for this work.

**Competing interests:** The authors have declared that no competing interests exist.

moderate-income increased risk, while a higher wealth index and stronger social support provided some protection.

## Conclusion

The findings from Woldia reveal a severe mental health crisis among women post-conflict, with elevated levels of depression, anxiety, and PTSD far exceeding global averages. This crisis jeopardizes the well-being of women and has far-reaching implications for families and communities, necessitating an urgent and multi-dimensional approach to address risk and protective factors identified in the study.

## Introduction

In developing countries, common mental health problems, comprising depression, anxiety disorders, and posttraumatic stress disorders, remain some of the most common and debilitating mental health problems, having severe public health, social, and economic impacts adversely affecting individuals' quality of life [1]. The triggers and manifestations of depression in these settings often intertwine with the unique socioeconomic and cultural factors inherent to these nations [2]. Generalized Anxiety Disorder (GAD) and Posttraumatic Stress Disorder (PTSD) are also prevalent mental health concerns in developing countries [3]. On the other hand, PTSD often stems from direct exposure to traumatic events, which can be all too common in developing countries due to conflicts, wars, natural disasters, or personal traumas like assault and abuse [4].

The global burden of depression, generalized anxiety disorders (GAD), and posttraumatic stress disorders (PTSD) is substantial and has profound implications for individuals, communities, and health systems worldwide [1]. In many parts of the world, especially in low-resource settings, the limited availability of mental health services, combined with societal stigma, including wartime conflicts, exacerbates the impact of these disorders, highlighting the urgent need for global strategies and interventions to address this mounting challenge [5].

In developing countries, several key determinants influence the mental health landscape. Pervasive poverty and associated challenges, such as food insecurity and limited access to necessities, often form the bedrock for mental distress [6]. The constant strains of political instability, civil unrest, or warfare create environments where trauma and subsequent mental disorders like PTSD [7]. Cultural and societal stigmas related to mental health often hinder individuals from seeking timely help, while the shortage of mental health professionals and infrastructure makes access to care challenging [8].

On top of these mental health stressors in developing countries, wartime has a severe negative impact on mental health, particularly for women. The mechanisms through which wartime exacerbates mental health issues in these women are multi-faceted, but to mention some, direct exposure to violence in which women witness or become victims of brutal acts of violence. Witnessing such violence can

lead to trauma, manifesting as PTSD or other stress-related disorders [9,10]. Conflict situations often involve a heightened risk of sexual assault, rape, and other gender-based violence. These traumatic events have significant psychological repercussions, leading to feelings of shame, guilt, depression, anxiety, and PTSD [9]. Many women lose family members due to direct conflict or related factors like disease or starvation. The grief and trauma associated with such loss can be overwhelming [11]. War often forces populations to flee their homes, leading to internal displacement or refuge in other countries. This dislocation from familiar settings and the hardships of refugee life can lead to feelings of rootlessness, anxiety, and depression [12].

Wartime often disrupts regular means of livelihood. Women, especially those heading households, face the pressure of providing for their families under constrained circumstances, leading to chronic stress. The traditional roles and societal structures can be upended during the war, leading to familial and societal expectations changes, which can further add to women's stress and mental health challenges [13]. Already constrained healthcare resources in developing countries become even more limited during the conflict, making it challenging to address physical and mental health needs. Women of reproductive age or those with children face added challenges, as war trauma can complicate pregnancies. In contrast, the immense stress of ensuring a child's safety and well-being in such an environment, coupled with the lasting impact of traumatic memories and the socioeconomic repercussions that persist long after the active conflict ends [14]. Such challenges can contribute to chronic mental health issues such as depression or PTSD, which may hinder a mother's ability to interact with her child, increasing the risk of neglect or even violent behavior, thereby perpetuating trauma across generations [15].

In the North Ethiopian conflict context, understanding the ripple effects of war on the population, especially women, is vital. War often creates an environment where social norms and structures are disrupted, leading to increased incidents like common mental health disorders. Furthermore, the traumatic experiences from such conditions, compounded by the general stresses of war, can significantly impact mental health. Women, already vulnerable due to the societal dynamics of the conflict region, may find themselves grappling with a range of mental health challenges as a direct outcome. Research in this area sheds light on the immediate effects but also aids in understanding the intricate relationship between war-induced incidents and mental health repercussions and may also support recommendations for targeted actions. Therefore, this research aims to assess the magnitude of war-related common mental health problems, depression, anxiety disorder, phobia, and posttraumatic stress disorder among women in Woldia district, Amhara, Ethiopia.

## Methods

**Setting**: The study took place in Woldia woreda, encompassing ten kebeles that experienced severe disruption, leading to significant humanitarian challenges for three months in 2021, with limited access to food, clean water, and sanitation. Woldia is a town and the capital of the North Wollo Zone in northern Ethiopia [16]. It is located 525 km north of Addis Ababa and 360 km from Bahir Dar, the Amhara Regional State's capital, north of Dessie and southeast of Lalibela. According to the 2007 Ethiopia national census, the projected population size of the town was 79,667 in 2010, with 49.4% being women [17]. The health institution profile of the city includes one public hospital, two public health centers, and several private pharmacies and clinics providing healthcare services to the community [16]. Woldia was occupied by the Tigray Defense Force (TDF) for three months before one year of this data collection. There was a heavy war between the TDF and the federal government forces, and they stayed for weeks at the entrance and later while the TDF left. During the three months of control by TDF, there was no definite law; people were displaced from their sites, heard of war situations, death of individuals, and closure of infrastructures like health facilities, schools, transportation, etc., which resulted in a reduction of individual movements.

Study Design: The study used a cross-sectional survey to measure conflict-related common mental disorders among women in the Woldia district from February to March 2023. The study included women in North Wollo zones of Amhara, where the conflict escalated, specifically women of 18 or more years who were in place during the war, living in Woldia town or a rural community, excluding women who were unable to communicate, chronically sick, and suffering from pain.

*Sample size and sampling:* We determined the sample size for the study based on previous evidence suggesting that 4.8% of women of reproductive age without IPV would have common mental disorders, primarily depression [18]. Assuming that women in conflict-affected areas with a lack of medication, social support, and exposure to IPV or GBV would experience a two-fold increase, we used a 95% CI, 80% power, and a design effect of 1.5. After adding 10% to compensate for possible non-response, we included 1,505 women in the study. We employed multistage sampling for the survey. We selected five of the ten available clusters in the first phase using simple random sampling. In the second stage, we chose households through systematic sampling, distributing the sample size to each kebele proportionate to the population size of households, taking households as the sampling unit. We created a sampling frame for each household with at least one eligible woman, assigning sequential numbers in a map sketch. After determining the sampling interval, data collectors randomly identified the first household as a starting point. They then added the sampling interval as a fixed increment, continuing until all chosen study subjects were interviewed. Enumerators randomly selected only one eligible woman for an interview in households with two or more eligible women.

*Data Collection and Measurement*: The study used a validated and standard questionnaire to measure socio-demographics, life experiences during wartime, and common mental illnesses. We developed the questionnaire in the local language, adapting it from previous versions of the instruments tested in other research. The study recruited and trained eighteen data collectors, two supervisors, and a coordinator for three days on the contents of the questionnaire. We pre-tested the questionnaire during the training of data collectors to amend any mismatches in local contextual linguistics. Enumerators collected the data using Open Data Kit, and we exported it into SPSS for Windows for cleaning and analysis.

The study included sociodemographic characteristics of the women, such as age, sex, current marital status, Oslo-social support index, life experience during wartime, and experiences related to threatening conditions. We measured the psychosocial problems of women using standard tools, including PHQ-9, the General Anxiety Disorder-7 (GAD-7), and the PTSD-Checklist for DSM-5 (PCL-5). The study also used the WHO-5 Well-being Index to supplement the women's mental health status assessment.

The PHQ-9 is a nine-item questionnaire measuring symptoms of depression. It is a well-established 9-item screening tool to assess symptoms of depression [19]. The module asks respondents to indicate how frequently they experienced depressive symptoms in the past two weeks, rating these on a scale of zero (never) to three (every day). The PHQ-9 is scored by summing up the nine responses, leading to a scale ranging between 0–27. As a severity measure, 5, 10, 15, and 20 cutoff points represent the thresholds for mild, moderate, moderately severe, and severe depression, respectively. If a single screening cutoff point is required, it is currently recommended to use a PHQ-9 score of 10 or greater, which has a high sensitivity and specificity of major depression [19].

The study also used the Generalized Anxiety Disorder (GAD-7), a seven-item tool assessing generalized anxiety disorder severity. Respondents rate symptoms from the past two weeks, with scores ranging from 0 to 3 based on frequency. It is a screening and severity measure for primary anxiety disorders, including GAD, Panic Disorder, Social Phobia, and PTSD. Validated in various settings, a study among Ethiopian University students [20] categorized GAD-7 scores: 0–4 as none, 5–9 as mild, 10–14 as moderate, and 15+ as severe anxiety [21]. Additionally, a single question evaluated functional difficulty related to anxiety.

Furthermore, the PTSD-PCL-5 is a 20-item measure that assesses the 20 DSM-5 symptoms of PTSD. The PCL-5 has a variety of purposes, including screening individuals for PTSD and making a provisional PTSD diagnosis. A total symptom severity score ranges from 0–80 and can be obtained by summing the scores for each of the twenty items. Research suggests a PCL-5 cutoff score between 31 and 33 indicates probable PTSD across samples. A provisional PTSD diagnosis can be made by treating each item rated as two "Moderately" or higher as a symptom endorsed, then following the *DSM-5* diagnostic rule, which requires at least 1 B item (questions 1–5), 1 C item (questions 6–7), 2 D items (questions 8–14), and 2 E items (questions 15–20) [22].

The WHO-5 Well-being is a five-item questionnaire that measures the general well-being of an individual in terms of a short form of quality of life. It has five items with six six-point Likert scales, ranging from '0 = total absence' to '5 = strong feeling' of well-being. The algorithm of the instrument includes summing the total score and multiplying the result by four to obtain a percentage score. The higher the score, the better an individual's well-being. The WHO-5 well-being questionnaire was used and validated in Ethiopia among adults to assess perceived overall change [23].

The study also assessed the Oslo-social support index (scale), which measures perceived social support and is used to evaluate the social well-being of participants. The Oslo Social Support Scale (OSSS-3) is assessed using a three-question survey, and each question is typically rated on a scale from 1 to 5, leading to a total score ranging from 3 to 14. The higher scores reflect more substantial perceived social support, with the following classifications: 3–8 (low), 9–11 (moderate), and 12–14 (high) social support [24].

## Possible confounding and effect-modifying variables

Possible confounding variables in this analysis include a range of women's social, economic, and war-related experiences and the substance use characteristics of their spouses. Key confounding variables that were measured include participants' age and sex, social status, household income, social networks, women's experience in the war, and spousal substance use. Furthermore, the study measured study subjects' socio-emotional competencies and coping skills using general self-efficacy and the brief COPE score. We examined the impact of sociodemographic and other factors that may interact with various effect-modifying variables.

## Data analysis

Data from the field was entered into an Excel spreadsheet on a computer and exported into SPSS for Windows. After cleaning and transforming them into meaningful variables, we assessed them for simple descriptive analysis. We further analyzed war-inflicted common mental health problems descriptively to find their magnitude in the sample and the population using a point estimate of the proportion and its 95% confidence interval, respectively.

The study used logistic regression to assess the association between socio-demographics, war-related disruptions, and common mental health problems. Each possible explanatory variable, including sociodemographic characteristics, war-related experiences, and other factors, was assessed for association first with each common mental illness status outcome variable, including depression, anxiety, and PTSD. We also evaluated each explanatory variable for association with the primary explanatory variable, war-related experiences. Explanatory variables having an association or borderline association with both the outcome and the explanatory variables were included in the analysis model between explanatory and outcome variables after assessing for multicollinearity.

## Ethical consideration

The study received ethical approval from the Institutional Review Board (IRB) of the Ethiopian Public Health Association, with a reference number EPHA/ OG/ 279/ 23 given on January 05, 2023. The letter of ethical approval, supplemented by a description of the purpose and procedures of the survey, was used to get permission from the local community. Participation in the study was generally voluntary, and informed consent was obtained orally from each subject. The study, as part of research on intimate partner violence, has used proper privacy and confidentiality as in the WHO guide for collecting data on violence against women [25]. Based on the guideline, consent was obtained orally from each participant after informing them of the study's objectives and the questionnaire's content, with details of the long-term possible benefits, including allowing for potential withdrawal from the study at any time. They were also informed that their withdrawal or participation would not be linked to any service use. Interviews were conducted only in a private setting and were not performed in the presence of any other individual over the age of 2 years. Strict confidentiality and privacy were ensured

throughout the study. Individuals with severe mental illness, severe depression, and severe forms of posttraumatic disorder, which the ODK software was notifying the enumerator, were referred to the local health facility after discussing with their supervisor. Women experiencing GBV were further informed about local resources related to gender issues. Moreover, the enumerators had a weekly debriefing to relieve the psychological secondary trauma from hearing about the violence experienced while interviewing.

## Results

The research encompassed 1,505 female participants from five kebeles within Woldia city. Most participants (41.7%) were aged between 25 and 34 years. The mean age of participants was 36.2 years, with an age range of 18–60 years. Over two-thirds of these participants were in a marital union. Regarding educational background, about 25% of the participants reported no formal education, while approximately 10% attained tertiary-level education. A significant 87.2% were religiously identified as Orthodox Christian adherents and 10.7% as Muslims. A predominant portion of these women was engaged in non-remunerative roles. In contrast, approximately 10% of the minor segment was employed in governmental or private sectors, "Table 1."

Economically, over 37% of participants reported a monthly income below 1,500 Ethiopian Birr. Two-thirds earned less than 3,000 Ethiopian Birr (approximately $60). When categorizing participants based on household assets, 26.5% were

**Table 1. Sociodemographic characteristics of women who were in Woldia during the wartime, Woldia, north Ethiopia, April 2023.**

| Characteristics | Frequency | Percent |
| --- | --- | --- |
| Age group | | |
| 18–24 yrs | 108 | 7.2 |
| 25–34 yrs | 623 | 41.4 |
| 35–44 yrs | 418 | 27.8 |
| 45–54 yrs | 235 | 15.0 |
| 55 yrs or more | 131 | 8.7 |
| Mean + SD | 36.2 + 10.2 | |
| Marital status | | |
| Married | 1087 | 72.2 |
| Currently single | 51 | 3.4 |
| Divorced or separated. | 215 | 14.3 |
| Widowed | 152 | 10.1 |
| Educational status | | |
| Not educated | 375 | 24.9 |
| Elementary [0–8] | 529 | 35.1 |
| Secondary [9 –12 ] | 453 | 30.1 |
| Tertiary [BSc or more] | 148 | 9.8 |
| Religion | | |
| Orthodox | 1312 | 87.2 |
| Muslim | 161 | 10.7 |
| Protestant | 32 | 2,1 |
| Occupation | | |
| Unpaid work | 1038 | 79.3 |
| Employee | 138 | 10.5 |
| Trading | 111 | 8.5 |
| Other. | 22 | 1.7 |

classified within the lower wealth index, while half belonged to the moderate category. Regarding self-perceived economic status relative to their neighbors, approximately 55.0% perceived themselves as in a lower bracket, and 39.3% viewed themselves as moderately positioned. Concerning community social interactions, 40.9% and 51.0% of participants perceived themselves as having a lower and moderate social status, respectively. Utilizing the Oslo-social support index as a metric, about half of the participants scored in the lower range, whereas about 15% achieved a strong score. In addition to income and social status, other familial factors also played a significant role in the participants' economic well-being, with some facing challenges due to spouses' substance use. About 11.0% of participants faced economic challenges attributed to their spouses' alcohol consumption, and about 4% encountered economic issues linked to their spouses' Khat chewing habits, "Table 2."

The magnitude of common mental health problems in the women population in Woldia during the war was evaluated as follows. The results indicate that 60.9% [95% CI: 58.5, 63.4] of women felt their lives lacked meaning. The magnitude of common mental health problems in the women population in Woldia post-wartime was 37.2% [95% CI: 34.7, 39.6] for any form of depression, 33.2% [95% CI: 30.8, 35.5] for any form of GAD, and 44.3% [95% CI: 38.0, 50.6] for PTSD. Considering the three common mental health problems, 50.1% [95% CI: 43.7, 56.5] have at least one form of the illness, while 33.0% [95% CI: 27.3, 38.6] have two or more of the above disorders.

**Table 2. Current reproductive health, Social, and economic related characteristics of women who were in Woldia during the wartime, Woldia, north Ethiopia, April 2023.**

| Characteristics | frequency | Percent |
|---|---|---|
| Monthly income [eth. Birr] | | |
| Below 1500 Birr | 564 | 37.7 |
| 1500–2999 Birr | 373 | 24.9 |
| Three thousand Birr or more. | 560 | 37.4 |
| Wealth index | | |
| Lower wealth | 404 | 26.8 |
| Medium wealth | 753 | 50.0 |
| Higher wealth. | 348 | 23.1 |
| Perceived economic status. | | |
| Higher | 84 | 5.6 |
| Moderate | 591 | 39.3 |
| Lower | 830 | 55.1 |
| Perceived social status. | | |
| Higher | 121 | 8.0 |
| Moderate | 768 | 51.0 |
| Lower | 616 | 40.9 |
| Oslo-social support index | | |
| Poor | 734 | 48.8 |
| Moderate | 547 | 36.3 |
| Strong | 223 | 14.8 |
| Spousal alcohol-related problem | | |
| No | 1340 | 89.0 |
| Yes | 165 | 11.0 |
| Spousal Khat-related problem | | |
| No | 1444 | 95.9 |
| Yes | 61 | 4.1 |

When assessing depression status individually, about 32.2% [95% CI: 29.8, 34.5] exhibited moderate to severe depressive symptoms, and 5.0% [95% CI: 3.9, 6.1] presented with severe forms of depressive manifestations. When measured against the criteria for generalized anxiety disorder (GAD), 22.7% [95% CI: 20.5, 24.8] demonstrated to have symptoms consistent with moderate GAD, while 10.5% [95% CI: 8.9, 12.0] met the criteria for a severe form of GAD. In terms of posttraumatic stress disorder (PTSD) symptomatology, 44.3% [95% CI: 41.8, 46.8] of women exhibited to have symptoms of the syndrome. Notably, within this segment, 20.9% displayed one or two major PTSD symptoms, and 23.4% manifested three or four major symptoms of the syndrome. Lastly, more than half of the women displayed at least one of the common mental health disorders mentioned above, of which 33.0% [95% CI: 30.5, 35.3] of the women exhibited a co-occurrence of two or more of these mental disorders, "Table 3."

## Determinants of common mental illness

The study broadly indicated that elements like marital status, the mother's job, family income, perceived social standing, and issues related to a spouse's use of alcohol and khat were commonly associated with depression, generalized anxiety disorder, and posttraumatic stress disorder. These factors were integrated into the models for these prevalent mental health issues. Moreover, age and perceived economic status had a specific association with depression. Religion, age at

**Table 3. The magnitude of common mental health problems among women who were in Woldia during the wartime, Woldia, north Ethiopia, April 2023.**

| Characteristics | Frequency | Percent (95% CI) |
|---|---|---|
| WHO-wellbeing [lower score] | 917 | 60.9% [58.5, 63.4] |
| PHQ-9 based scoring. | | |
| None | | |
| Moderate [10 –19 ] | 484 | 32.2 [29.8, 34.5] |
| Severe depression [20 or more] | 75 | 5.0 [3.9, 6.1] |
| Moderate to severe [10-to 27] | 559 | 37.2 [34.7, 39.6] |
| Generalized anxiety disorder | | |
| None | | |
| Moderate anxiety | 341 | 22.7 [20.5, 24.8] |
| Severe anxiety | 158 | 10.5 [8.9, 12.0] |
| Moderate to severe | 499 | 33.2 [30.8, 35.5] |
| Presence of PTSD | | |
| None | | |
| Present | 667 | 44.3 [41.8, 46.8] |
| Symptoms of PTSD | | |
| Intrusion symptoms | 568 | 37.7 [35.3, 40.2] |
| Avoidance of stimuli | 520 | 34.6 [32.1, 37.0] |
| Alteration of cognition and mood | 298 | 19.8 [17.8, 21.8] |
| Alteration in arousal and reactivity | 315 | 20.9 [18.9, 23.0] |
| Number of Symptoms of PTSD | | |
| 1–2 major symptoms | 315 | 20.9 [19.9, 23.0] |
| 3–4 major symptoms | 352 | 23.4 [21.3, 25.5] |
| Comorbidity | | |
| None | 751 | 49.9 [47.4, 52.4] |
| Only one disorder | 258 | 17.1 [15.2, 19.0] |
| Two or more disorders | 496 | 33.0 [30.5, 35.3] |
| Any form of mental illness | 754 | 50.1 [47.6, 52.6] |

the time of marriage, and perceived economic status were related to generalized anxiety disorder. A woman's number of children was explicitly linked to posttraumatic stress disorder. These specialized factors were also included in the respective models for each mental health condition, "Tables 4 and 5."

After accounting for various factors, we found that depressive symptoms were more common among older women, single women, and women whose spouses have issues related to khat chewing. On the other hand, women in trading jobs, those with a medium to high wealth index, and those who perceive themselves as having moderate to high social and economic status, as well as those with moderate to strong social support (measured by the Oslo-social support index), were less likely to report depression.

In the case of generalized anxiety disorder, older age, single marital status, and moderate monthly income were positively associated with the condition. Conversely, having a medium to high wealth index and moderate to strong social support (again, measured by the Oslo-social support index) reduced the likelihood of experiencing generalized anxiety disorder.

**Table 4. Sociodemographic correlates of common mental health problems of women who were in Woldia during the wartime, Woldia, north Ethiopia, April 2023.**

| Characteristics | Sample | Depression (PHQ-9 ≥ 10) n (%) | Crude OR (95% CI) | Anxiety Dis. (Mod-Severe) n (%) | Crude OR (95% CI) | PTSD n (%) | Crude OR (95% CI) |
|---|---|---|---|---|---|---|---|
| Age group | | | | | | | |
| 18–24 yrs | 108 | 32 (29.6) | 1.00 | 35 (32.4) | 1.00 | 37 (34.3) | 1.00 |
| 25–34 yrs | 623 | 222 (35.6) | 1.32 (0.84, 2.05) | 200 (32.1) | 0.99 (0.64, 1.52) | 186 (29.9) | 0.82 (0.53, 1.26) |
| 35–44 yrs | 418 | 152 (36.4) | 1.36 (0.86, 2.15) | 138 (33.0) | 1.03 (0.65, 1.62) | 150 (35.9) | 1.07 (0.69, 1.68) |
| 45–54 yrs | 235 | 92 (40.9) | 1.64 (1.01, 2.69) | 76 (33.8) | 1.06 (0.65, 1.73) | 81 (36.0) | 1.08 (0.67, 1.75) |
| 55 yrs or more | 131 | 61 (46.6) | 2.07 (1.21, 3.54) | 50 (38.2) | 1.29 (0.75, 2.20) | 54 (41.2) | 1.35 (0.79, 2.28) |
| Marital status | | | | | | | |
| Married | 1087 | 366 (33.7) | 1.00 | 340 (31.3) | 1.00 | 326 (30.0) | 1.00 |
| Currently single | 51 | 29 (56.9) | 2.59 (1.47, 4.58) | 28 (54.6) | 2.68 (1.52, 4.71) | 38 (74.5) | 6.82 (3.59, 12.9) |
| Divorced or separated. | 215 | 95 (44.2) | 1.56 (1.16, 2.10) | 75 (34.9) | 1.18 (0.87, 1.60) | 75 (34.9) | 1.25 (0.92, 1.70) |
| Widowed | 152 | 69 (45.4) | 1.64 (1.16, 2.31) | 56 (36.8) | 1.28 (0.90, 1.83) | 69 (45.4) | 1.94 (1.38, 2.74) |
| Educational status | | | | | | | |
| Not educated | 375 | 170 (45.3) | 1.00 | 133 (35.5) | 1.00 | 146 (38.9) | 1.00 |
| Elementary [0–8] | 529 | 194 (36.7) | 0.70 (0.53, 0.91) | 193 (36.5) | 1.05 (0.79, 1.38) | 180 (34.0) | 0.81 (0.62, 1.07) |
| Secondary [9 –12 ] | 453 | 161 (35.5) | 0.67 (0.50, 0.88) | 140 (30.9) | 0.81 (0.61, 1.09) | 146 (32.2) | 0.75 (0.56, 0.99) |
| Tertiary [BSc or more] | 148 | 34 (23.0) | 0.36 (0.23, 0.56) | 33 (22.3) | 0.52 (0.34, 0.81) | 36 (24.3) | 0.50 (0.33, 0.77) |
| Religion | | | | | | | |
| Orthodox | 1312 | 496 (37.8) | 1.00 | 448 (34.1) | 1.00 | 441 (33.6) | 1.00 |
| Muslim | 161 | 52 (32.3) | 0.79 (0.55, 1.11) | 41 (25.5) | 0.66 (0.45, 0.96) | 56 (34.8) | 1.05 (0.75, 1.49) |
| Protestant | 32 | 11 (34.4) | 0.86 (0.41, 1.80) | 10 (31.3) | 0.88 (0.41, 1.87) | 11 (34.4) | 1.04 (0.49, 2.17) |
| Occupation | | | | | | | |
| Unpaid work | 1038 | 422 (40.7) | 1.00 | 370 (35.6) | 1.00 | 383 (36.9) | 1.00 |
| Employee | 138 | 30 (21.7) | 0.41 (0.27, 0.62) | 26 (18.8) | 0.42 (0.27, 0.65) | 32 (23.2) | 0.52 (0.34, 0.78) |
| Trading | 111 | 30 (27.0) | 0.54 (0.35, 0.84) | 32 (28.8) | 0.73 (0.48, 1.12) | 28 (25.2) | 0.58 (0.37, 0.90) |
| Other. | 22 | 14 (63.6) | 2.56 (1.06, 6.14) | 11 (50.0) | 1.81 (0.78, 4.20) | 12 (54.5) | 2.05 (0.88, 4.80) |
| Monthly income [eth. Birr] | | | | | | | |
| Below 1500 Birr | 564 | 222 (39.4) | 1.00 | 191 (33.9) | 1.00 | 207 (36.7) | 1.00 |
| 1500–2999 Birr | 373 | 185 (49.6) | 1.52 (1.16, 1.97) | 183 (49.1) | 1.88 (1.44, 2.46) | 175 (46.9) | 1.52 (1.17, 1.99) |
| 3000 Birr or more. | 560 | 148 (26.4) | 0.55 (0.43, 0.71) | 125 (22.3) | 0.56 (0.43, 0.73) | 124 (22.1) | 0.49 (0.38, 0.64) |

**Table 5. Current reproductive health, Social, and economic related correlates of common mental health problem of women who were in Woldia during the wartime, Woldia, north Ethiopia, April 2023.**

| Characteristics | Sample | Depression (PHQ-9 ≥ 10) n (%) | Crude OR (95% CI) | Anxiety Dis. (Mod-severe) n (%) | Crude OR (95% CI) | PTSD n (%) | Crude OR (95% CI) |
|---|---|---|---|---|---|---|---|
| Age at marriage | | | | | | | |
| < 15 yrs | 97 | 28 (28.9) | 1.00 | 23 (23.7) | 1.00 | 34 (35.1) | 1.00 |
| 15–17 yrs | 317 | 123 (38.8) | 1.56 (0.95, 2.56) | 99 (31.2) | 1.46 (0.87, 2.47) | 106 (33.4) | 0.93 (0.58, 1.50) |
| 18 yrs. or more. | 1070 | 397 (37.1) | 1.45 (0.92, 2.29) | 369 (34.5) | 1.69 (1.04, 2.75) | 359 (33.6) | 0.94 (0.61, 1.45) |
| Number of children | | | | | | | |
| No child | 117 | 47 (40.2) | 1.00 | 41 (35.0) | 1.00 | 54 (46.2) | 1.00 |
| 1–2 children | 849 | 315 (37.1) | 0.88 (0.59, 1.30) | 282 (33.2) | 0.92 (0.61, 1.38) | 284 (33.5) | 0.59 (0.40, 0.87) |
| 3 or more | 539 | 197 (36.5) | 0.86 (0.57, 1.29) | 176 (32.7) | 0.90 (0.59, 1.37) | 170 (31.5) | 0.54 (0.36, 0.81) |
| Wealth index | | | | | | | |
| Lower wealth | 404 | 231 (57.2) | 1.00 | 213 (52.7) | 1.00 | 196 (48.5) | 1.00 |
| Medium wealth | 753 | 252 (33.5) | 0.38 (0.29, 0.48) | 222 (29.5) | 0.38 (0.29, 0.48) | 238 (31.7) | 0.49 (0.39, 0.83) |
| Higher wealth | 348 | 76 (21.8) | 0.21 (0.15, 0.29) | 64 (18.4) | 0.20 (0.15, 0.28) | 73 (21.0) | 0.62 (0.49, 0.77) |
| Perceived economic status. | | | | | | | |
| Higher | 84 | 15 (17.9) | 0.28 (0.16, 0.50) | 12 (14.3) | 0.27 (0.14, 0.50) | 20 (23.8) | 0.49 (0.29, 2.33) |
| Moderate | 591 | 182 (30.8) | 0.56 (0.46, 0.72) | 168 (28.4) | 0.64 (0.51, 0.80) | 166 (28.1) | 0.74 (0.50, 1.08) |
| Lower | 830 | 362 (43.6) | 1.00 | 319 (38.4) | 1.00 | 322 (38.8) | 1.00 |
| Perceived social status. | | | | | | | |
| Higher | 121 | 26 (21.5) | 0.29 (0.19, 0.47) | 23 (19.0) | 0.33 (0.20, 0.53) | 33 (27.3) | 0.47 (0.30, 0.72) |
| Moderate | 768 | 236 (30.7) | 0.48 (0.38, 0.59) | 219 (28.5) | 0.56 (0.45, 0.70) | 200 (26.0) | 0.44 (0.35, 0.55) |
| Lower | 616 | 297 (48.2) | 1.00 | 257 (41.7) | 1.00 | 275 (44.6) | 1.00 |
| Oslo-social support index | | | | | | | |
| Poor | 734 | 347 (47.3) | 1.00 | 301 (41.0) | 1.00 | 274 (37.3) | 1.00 |
| Moderate | 547 | 173 (31.6) | 0.52 (0.41, 0.65) | 162 (29.6) | 0.61 (0.48, 0.77) | 188 (34.4) | 0.88 (0.70, 1.11) |
| Strong | 223 | 39 (17.5) | 0.24 (0.16, 0.34) | 36 (16.1) | 0.28 (0.19, 0.41) | 45 (20.2) | 0.42 (0.30, 0.61) |
| Spousal alcohol-related prob. | | | | | | | |
| No | 1340 | 478 (35.7) | 1.00 | 428 (31.9) | 1.00 | 448 (33.4) | 1.00 |
| Yes | 165 | 81 (49.1) | 1.74 (1.26, 2.41) | 71 (43.0) | 1.61 (1.16, 2.24) | 60 (36.4) | 1.14 (0.81, 1.59) |
| Spousal Khat-related problem | | | | | | | |
| No | 1444 | 527 (36.5) | 1.00 | 471 (32.6) | 1.00 | 483 (33.4) | 1.00 |
| Yes | 61 | 32 (52.5) | 1.92 (1.15, 3.21) | 28 (45.9) | 1.75 (1.05, 2.94) | 25 (41.0) | 1.38 (0.82, 2.33) |

Lastly, for posttraumatic stress disorder (PTSD), the condition was more common among older women, those who are currently single or widowed, women with moderate monthly incomes, and women with moderate levels of social support. However, women in trading occupations, those with a medium to high wealth index, those who perceive their social status as moderate, and those with strong social support were less likely to have PTSD, "Tables 6 and 7."

## Discussion

In Woldia, one year after wartime, the study found alarming rates of common mental health problems among women. Sixty-one percent felt life was a problematic well-being consequence, while 37.2% had any form of depression, [32.2% had moderate to severe and 5% had severe depression.]. Similarly, 33.2% [22.7% had moderate and 10.5% had severe]

**Table 6. Sociodemographic correlates of common mental health problems of women who were in Woldia during the wartime, Woldia, north Ethiopia, April 2023.**

| Characteristics | Depression [PHQ-9 score ≥ 10] | | Generalized Anxiety Disorder (GAD) [Moderate or Severe GAD] | | Posttraumatic stress disorder [PTSD] | |
|---|---|---|---|---|---|---|
| | Crude OR (95% CI) | Adjusted* OR (95% CI) | Crude OR (95% CI) | Adjusted** OR (95% CI) | Crude OR (95% CI) | Adjusted*** OR (95% CI) |
| Age group | | | | | | |
| 18–24 yrs | 1.00 | 1.00 | 1.00 | 1.00 | 1.00 | 1.00 |
| 25–34 yrs | 1.32 (0.84, 2.05) | 2.03 (1.22, 3.39) | 0.99 (0.64, 1.52) | 1.24 (0.77, 1.99) | 0.82 (0.53, 1.26) | 1.13 (0.68, 1.88) |
| 35–44 yrs | 1.36 (0.86, 2.15) | 2.44 (1.41, 4.25) | 1.03 (0.65, 1.62) | 1.51 (0.92, 2.50) | 1.07 (0.69, 1.68) | 2.10 (1.21, 3.66) |
| 45–54 yrs | 1.64 (1.01, 2.69) | 3.51 (1.90, 6.47) | 1.06 (0.65, 1.73) | 1.71 (0.98, 3.00) | 1.08 (0.67, 1.75) | 2.40 (1.29, 4.47) |
| 55 yrs or more | 2.07 (1.21, 3.54) | 3.10 (1.56, 6.16) | 1.29 (0.75, 2.20) | 2.32 (1.22, 4.41) | 1.35 (0.79, 2.28) | 2.03 (1.01, 4.07) |
| Marital status | | | | | | |
| Married | 1.00 | 1.00 | 1.00 | 1.00 | 1.00 | 1.00 |
| Currently single | 2.59 (1.47, 4.58) | 2.39 (1.18, 4.83) | 2.68 (1.52, 4.71) | 2.36(1.16, 4.78) | 6.82 (3.59, 12.9) | 7.35 (3.14, 17.2) |
| Divorced or separated. | 1.56 (1.16, 2.10) | 1.17 (0.81, 1.69) | 1.18 (0.87, 1.60) | 0.95 (0.65, 1.39) | 1.25 (0.92, 1.70) | 0.90 (0.62, 1.30) |
| Widowed | 1.64 (1.16, 2.31) | 1.35 (0.86, 2.11) | 1.28 (0.90, 1.83) | 1.49 (0.96, 2.31) | 1.94 (1.38, 2.74) | 1.68 (1.08, 2.59) |
| Educational status | | | | | | |
| Not educated | 1.00 | 1.00 | 1.00 | 1.00 | 1.00 | 1.00 |
| Elementary [0–8] | 0.70 (0.53, 0.91) | 0.97 (0.69, 1.37) | 1.05 (0.79, 1.38) | 1.10 (0.79, 1.52) | 0.81 (0.62, 1.07) | 1.09 (0.51, 1.44) |
| Secondary [9 –12 ] | 0.67 (0.50, 0.88) | 1.03 (0.70, 1.51) | 0.81 (0.61, 1.09) | 0.92 (0.64, 1.34) | 0.75 (0.56, 0.99) | 1.13 (0.76, 1.67) |
| Tertiary [BSc or more] | 0.36 (0.23, 0.56) | 0.89 (0.49, 1.59) | 0.52 (0.34, 0.81) | 0.80 (0.47, 1.34) | 0.50 (0.33, 0.77) | 0.88 (0.48, 1.60) |
| Occupation | | | | | | |
| Unpaid work | 1.00 | 1.00 | 1.00 | 1.00 | 1.00 | 1.00 |
| Employee | 0.77 (0.46, 1.28) | 0.77 (0.46, 1.28) | 0.42 (0.27, 0.65) | 1.05 (0.76, 1.47) | 0.52 (0.34, 0.78) | 0.86 (0.51, 1.44) |
| Trading | 0.52 (0.32, 0.84) | 0.52 (0.32, 0.84) | 0.73 (0.48, 1.12) | 0.82 (0.56, 1.20) | 0.58 (0.37, 0.90) | 0.61 (0.38, 0.99) |
| Other. | 2.56 (0.97, 6.71) | 2.56 (0.97, 6.71) | 1.81 (0.78, 4.20) | 0.72 (0.40, 1.29) | 2.05 (0.88, 4.80) | 2.16 (0.85, 5.48) |
| Religion | | | | | | |
| Orthodox | 1.00 | = = | 1.00 | 1.00 | 1.00 | = = |
| Muslim | 0.79 (0.55, 1.11) | ========= | 0.66 (0.45, 0.96) | 0.81 (0.52, 1.27) | 1.05 (0.75, 1.49) | ========= |
| Protestant | 0.86 (0.41, 1.80) | ========= | 0.88 (0.41, 1.87) | 1.47 (0.61, 3.51) | 1.04 (0.49, 2.17) | ========= |
| Monthly income [eth. Birr] | | | | | | |
| Below 1500 Birr | 1.00 | 1.00 | 1.00 | 1.00 | 1.00 | 1.00 |
| 1500–2999 Birr | 1.52 (1.16, 1.97) | 1.55 (1.15, 2.09) | 1.88 (1.44, 2.46) | 1.84 (1.36, 2.49) | 1.52 (1.17, 1.99) | 1.40 (1.04, 1.88) |
| 3000 Birr or more. | 0.55 (0.43, 0.71) | 1.07 (0.77, 1.48) | 0.56 (0.43, 0.73) | 0.91 (0.65, 1.28) | 0.49 (0.38, 0.64) | 0.80 (0.57, 1.11) |

*Depression: Age grouped, Occupational status, educational status, Marital status, Monthly income, Household wealth, Oslo social support status, perceived economic position, perceived social position, presence of alcohol drinking problem, Presence of Khat chewing-related problem.

**Variable(s) entered on step 1: Occupational status, educational status, Marital status, Monthly income, Household wealth, Oslo social support status, perceived social position, presence of alcohol drinking problem, Presence of Khat chewing related problem, age at marriage, Religion, Perceived economic position.

***Variable(s) entered on step 1: Occupational status, educational status, Marital status, number of children (biological), Monthly income, Household wealth, Oslo social support status, perceived economic position, perceived social position, age group.

forms of generalized anxiety disorders (GAD), and 44.3% exhibited symptoms of posttraumatic stress disorder (PTSD). Considering the three common mental health problems, 50.1% have at least one form of the illness, while 33.0% [95% CI: 27.3, 38.6] have two or more of the above disorders. Various risk and protective factors were identified: depressive symptoms were more prevalent among older, single women and those with spouses using khat, while women with higher wealth indices and strong social support were less likely to suffer. Similar patterns were observed for GAD and PTSD.

**Table 7. Current reproductive health, Social, and economic related correlates of common mental health problem of women who were in Woldia during the wartime, Woldia, north Ethiopia, April 2023.**

| Characteristics | Depression [PHQ-9 score ≥ 10] | | Generalized Anxiety Disorder (GAD) [Moderate or Sever GAD] | | Posttraumatic stress disorder [PTSD] | |
|---|---|---|---|---|---|---|
| | Crude OR (95% CI) | Adjusted* OR (95% CI) | Crude OR (95% CI) | Adjusted** OR (95% CI) | Crude OR (95% CI) | Adjusted*** OR (95% CI) |
| Age at marriage | | | | | | |
| <15 yrs | 1.00 | = = | 1.00 | 1.00 | 1.00 | = = |
| 15–17 yrs | 1.56 (0.95, 2.56) | ========= | 1.46 (0.87, 2.47) | 1.44 (0.81, 2.57) | 0.93 (0.58, 1.50) | ========= |
| 18 yrs. or more. | 1.45 (0.92, 2.29) | ========= | 1.69 (1.04, 2.75) | 1.70 (0.98, 2.94) | 0.94 (0.61, 1.45) | ========= |
| Number of children | | | | | | |
| No child | 1.00 | = = | 1.00 | = = | 1.00 | 1.00 |
| 1–2 Children | 0.88 (0.59, 1.30) | ========= | 0.92 (0.61, 1.38) | ========= | 0.59 (0.40, 0.87) | 0.89 (0.54, 1.48) |
| 3 or more | 0.86 (0.57, 1.29) | ========= | 0.90 (0.59, 1.37) | ========= | 0.54 (0.36, 0.81) | 0.67 (0.39, 1.17) |
| Wealth index | | | | | | |
| Lower wealth | 1.00 | 1.00 | 1.00 | 1.00 | 1.00 | 1.00 |
| Medium wealth | 0.38 (0.29, 0.48) | 0.45 (0.34, 0.60) | 0.38 (0.29, 0.48) | 0.40 (0.30, 0.54) | 0.49 (0.39, 0.83) | 0.58 (0.43, 0.78) |
| Higher wealth | 0.21 (0.15, 0.29) | 0.33 (0.21, 0.53) | 0.20 (0.15, 0.28) | 0.28 (0.17, 0.46) | 0.62 (0.49, 0.77) | 0.45 (0.28, 0.72) |
| Perceived economic status. | | | | | | |
| Higher | 0.28 (0.16, 0.50) | 0.94 (0.37, 2.40) | 0.27 (0.14, 0.50) | 0.74 (0.26, 2.14) | 0.49 (0.29, 2.33) | ========= |
| Moderate | 0.56 (0.46, 0.72) | 1.17 (0.82, 1.67) | 0.64 (0.51, 0.80) | 1.27 (0.88, 1.84) | 0.74 (0.50, 1.08) | ========= |
| Lower | 1.00 | 1.00 | 1.00 | 1.00 | 1.00 | = = |
| Perceived social status. | | | | | | |
| Higher | 0.29 (0.19, 0.47) | 0.70 (0.34, 1.44) | 0.33 (0.20, 0.53) | 0.89 (0.42, 1.92) | 0.47 (0.30, 0.72) | 1.00 (0.51, 1.97) |
| Moderate | 0.48 (0.38, 0.59) | 0.72 (0.52, 0.99) | 0.56 (0.45, 0.70) | 0.88 (0.63, 1.23) | 0.44 (0.35, 0.55) | 0.51 (0.36, 0.71) |
| Lower | 1.00 | 1.00 | 1.00 | 1.00 | 1.00 | 1.00 |
| Oslo-social support index | | | | | | |
| Poor | 1.00 | 1.00 | 1.00 | 1.00 | 1.00 | 1.00 |
| Moderate | 0.52 (0.41, 0.65) | 0.71 (0.54, 0.95) | 0.61 (0.48, 0.77) | 0.69 (0.53, 0.90) | 0.88 (0.70, 1.11) | 1.35 (1.01, 1.80) |
| Strong | 0.24 (0.16, 0.48) | 0.27 (0.17, 0.49) | 0.28 (0.19, 0.41) | 0.34 (0.22, 0.52) | 0.42 (0.30, 0.61) | 0.53 (0.33, 0.85) |
| Spousal alcohol-related prob. | | | | | | |
| No | 1.00 | 1.00 | 1.00 | 1.00 | 1.00 | = = |
| Yes | 1.74 (1.26, 2.41) | 1.42 (0.94, 2.16) | 1.61 (1.16, 2.24) | 1.36 (0.0.89, 2.07) | 1.14 (0.81, 1.59) | ========= |
| Spousal Khat-related prob. | | | | | | |
| No | 1.00 | 1.00 | 1.00 | 1.00 | 1.00 | = = |
| Yes | 1.92 (1.15, 3.21) | 2.41 (1.25, 4.65) | 1.75 (1.05, 2.94) | 1.84 (0.94, 3.57) | 1.38 (0.82, 2.33) | ========= |

*Depression: Age grouped, Occupational status, educational status, Marital status, Monthly income, Household wealth, Oslo social support status, perceived economic position, perceived social position, presence of alcohol drinking problem, Presence of Khat chewing-related problem.

**Variable(s) entered on step 1: Occupational status, educational status, Marital status, Monthly income, Household wealth, Oslo social support status, perceived social position, presence of alcohol drinking problem, Presence of Khat chewing related problem, age at marriage, Religion, Perceived economic position.

***Variable(s) entered on step 1: Occupational status, educational status, Marital status, number of children (biological), Monthly income, Household wealth, Oslo social support status, perceived economic position, perceived social position, age group.

Older age, single status, and moderate income were risk factors, while a higher wealth index and more substantial social support offered some protection.

The findings from Woldia one year after wartime present a disturbing picture of the mental health conditions among women in the area, with elevated rates of depression, generalized anxiety disorder (GAD), and posttraumatic stress

disorder (PTSD). The burden of common mental illnesses in post-conflict areas is significantly higher compared to populations in stable settings. Compared to global averages, these rates are exceedingly high. For instance, the World Health Organization estimates that, globally, around 4.4% of the population suffers from depression [26,27] and 3.6% from anxiety disorders (28). In contrast, our study in Woldia found that 37.2% of women showed signs of a moderate or severe form of depression, and 33.2% had moderate or severe generalized anxiety disorder. Compared to local research in the Amhara region, it was higher than the study done in three hospitals among mother caregivers of children or adolescents with epilepsy, with a prevalence of depression and anxiety of 13.7% and 10.4%, respectively [28]. It is still higher than research done among hypertensive and diabetic patients in Debrebirhan, having a prevalence depression of 19.5% [29]. Such drastic differences underscore the devastating mental health impact of wartime experiences on women in developing countries like Woldia, where post-war poverty is high and access to mental health services is often limited.

Furthermore, the prevalence of PTSD among women in Woldia, at 44.3%, also stands out as alarming when compared to other conflict and post-conflict regions. Research in other developing countries affected by conflict, such as Rwanda and Bosnia, has reported PTSD rates ranging from 20% to 40% [30], suggesting that the situation in Woldia is at the higher end of the distress scale. The high percentage of women who feel their life in misery stresses attending life in poverty and lacking a meaningful life (61%) is another concerning metric that points to existential crises, often less quantified but deeply impactful in terms of long-term well-being. The findings from Woldia, reflecting the mental health conditions of women in the aftermath of wartime, are deeply concerning. Almost half of the women exhibited significant symptoms associated with one of the three common mental health problems. This high prevalence rate highlights the profound psychological trauma that such conflicts can inflict on vulnerable populations, especially women in developing regions. Comparatively, in many global contexts outside of conflict zones, the prevalence of such mental health disorders is typically much lower. The added distress is attributed to the direct and indirect impacts of the war, including loss of loved ones, physical harm, displacement, and overarching social disruption.

Moreover, approximately 33% of these women demonstrated comorbidity – or the co-occurrence of two or more disorders – which underscores the complexity and depth of their mental health challenges. Comorbidity often suggests that affected individuals experience more severe psychological distress, significantly complicating treatment pathways. For instance, treating a person with depression alone differs from treating someone with depression and generalized anxiety disorder. The combined impact of comorbid disorders can lead to exacerbated symptoms, prolonged suffering, and increased disability. In other global contexts, particularly in developed nations, there is an established recognition of the challenges posed by comorbidity, with specific treatment regimens designed to address such complexities.

Given the grim picture painted by these findings, it becomes imperative to consider the broader implications for Woldia and similar regions. Women are foundational pillars in the fabric of society, often holding familial and community structures together. If their mental well-being is compromised, the ripple effect can impact families, children's development, community cohesion, and societal progress.

The relationship between sociodemographic factors and mental health outcomes among women in Woldia mirrors findings from other studies, affirming the global relevance of these connections. Previous research has demonstrated a higher prevalence of depressive symptoms among older individuals [31] and single women [32]. Similarly, spouse substance use, such as khat chewing, has been identified as a stressor contributing to mental health conditions in other contexts [33]. These risk factors apply consistently to generalized anxiety disorder (GAD) and posttraumatic stress disorder (PTSD), suggesting that age, marital status, and spousal behaviors are critical determinants across various mental health disorders. The effect of those sociodemographic conditions may worsen these common mental health problems due to lack of emotional and social support from others, while they are single, cumulative stress and trauma augmented by social isolation in older women during war time, and disruption of coping mechanism augmented by social stigma and isolation during war time of women whose spouses substance use could worsen common mental health problems.

The study also sheds light on protective factors, namely higher wealth indices and strong social support, associated with lower rates of depressive symptoms, GAD, and PTSD. These findings align with a body of literature emphasizing the importance of social capital and economic stability in mental health outcomes [34]. Social support, often measured through tools like the Oslo-social support index, has been shown to provide a buffer against the adverse psychological impacts of stress and trauma [35] and is particularly important in the context of developing nations where formal mental health services may be lacking. The protective effect of social and economic stability may be attributed to the buffering role of social support in mitigating psychological stressors. This stability also may function as social capital, fostering community resilience by enhancing security, psychosocial well-being, and economic support in the face of challenges arising during wartime conflicts.

The consistent association of risk and protective factors across the three major mental health issues of depression, GAD, and PTSD suggests that interventions to improve mental health in this context should adopt a multi-directional approach. Addressing root causes like substance abuse in spouses and providing targeted mental health services for older, single women could be critical steps. This gender disparity in depression is well known among older people, although the exact cause is not well known [36,37]. Similarly, single women are more vulnerable to such common mental illness [38]. Simultaneously, community-based programs that enhance social support networks and economic empowerment initiatives aimed at improving the wealth index of women can serve as effective preventative measures [39]. Given the multi-dimensional nature of these risk and protective factors, during wartime conflicts, vulnerable groups such as older adults, single individuals, and women with spouses engaged in substance use require targeted support to mitigate mental health risks. Establishing safe shelters, mobilizing community networks, and providing crisis support services can help reduce psychological distress. Strengthening protective factors, including social support networks and economic empowerment initiatives, further enhances resilience by fostering peer support, offering financial aid, and enabling self-sufficiency through skills training and integrating mobile mental health clinics, community counseling, and long-term recovery.

The study on mental health issues among women in Woldia after wartime provides crucial insights. Still, it has limitations regarding generalizability, temporal scope, and depth of analysis of protective factors. Specifically, focusing on a region deeply affected by conflict and post-war conditions may limit the study's applicability to other areas. The cross-sectional nature of the research provides a "snapshot" rather than a longitudinal view, potentially missing evolving trends in mental health. Lastly, while protective factors like wealth indices and social support are mentioned, a more in-depth exploration of these elements, such as social support types and quality and their longitudinal impact, could offer a more nuanced understanding of their role in mental health outcomes.

## Conclusion

The findings from Woldia reveal a severe mental health crisis among women in the aftermath of wartime, with elevated levels of depression, anxiety, and PTSD far exceeding global averages. This situation not only jeopardizes the well-being of women but also has far-reaching implications for families and communities. Addressing this crisis is urgent and requires a multi-pronged approach, considering both risk and protective factors identified in the study.

## Recommendations

Given the dire situation, immediate and sustained interventions are essential. Multi-disciplinary teams should be deployed to provide targeted mental health services, particularly for vulnerable groups like older and single women in war-affected places. Given the high prevalence of GBV and PTSD symptoms, we recommend trauma-informed care and psychotherapy, including community-based approaches and training healthcare workers and supporting staff to provide trauma-sensitive support. Furthermore, community-based initiatives should be implemented to strengthen social support networks and offer economic empowerment. Coordination between social, financial, and healthcare sectors is imperative for creating a holistic

support system that can address the complex challenges posed by high rates of mental disorders and comorbidities among women in Woldia and similar conflict-affected areas.

## Acknowledgments

We want to acknowledge the contributions of the following groups: the Woreda Health Office and the Woreda Women's and Children's Affairs Office of Woldia town for their positive consent and support in using district data for sampling. We extend our gratitude to the enumerators who collected the data in the field and the supervisors who diligently assisted and monitored the data collectors. Finally, we are deeply grateful to the women who participated and consented to share their real experiences.

## Author contributions

**Conceptualization:** Negussie Deyessa, Abraraw Tadesse.

**Data curation:** Kehali Getahun, Ruth Filimona.

**Formal analysis:** Negussie Deyessa, Kehali Getahun, Ruth Filimona, Abraraw Tadesse.

**Methodology:** Negussie Deyessa, Abraraw Tadesse.

**Visualization:** Negussie Deyessa, Kehali Getahun, Ruth Filimona, Abraraw Tadesse.

**Writing – original draft:** Negussie Deyessa.

**Writing – review & editing:** Negussie Deyessa, Kehali Getahun, Ruth Filimona, Abraraw Tadesse.

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
