## [Decision Letter · Decision Letter 0]

7 Jan 2025

PONE-D-24-23457The impact of wartime conflict on mental health problems of women in conflict-hit population in Woldia, EthiopiaPLOS ONE

Dear Dr. Deyessa,

Thank you for submitting your manuscript to PLOS ONE. After careful consideration, we feel that it has merit but does not fully meet PLOS ONE’s publication criteria as it currently stands. Therefore, we invite you to submit a revised version of the manuscript that addresses the points raised during the review process.

**Both reviewers highlighted the significance of your research and shared their enthusiasm for your contribution to the literature, which I echo. While their feedback included only minor suggestions, these are important for strengthening your manuscript. Before proceeding with acceptance, I kindly ask that you address each of the reviewers' comments and revise the relevant sections of your manuscript accordingly and provide a point-by-point response letter.**

We look forward to receiving your revised manuscript.

Kind regards,

Inga Schalinski

Academic Editor

PLOS ONE

**Journal Requirements:**

2. Please provide additional details regarding participant consent. In the ethics statement in the Methods and online submission information, please ensure that you have specified (a) whether consent was informed and (b) what type you obtained (for instance, written or verbal, and if verbal, how it was documented and witnessed). If your study included minors, state whether you obtained consent from parents or guardians. If the need for consent was waived by the ethics committee, please include this information.

3. We are unable to open your Supporting Information file "data, Mar 08.sav". Please kindly revise as necessary and re-upload.

Reviewers' comments:

Reviewer's Responses to Questions

**Comments to the Author**

1. Is the manuscript technically sound, and do the data support the conclusions?

Reviewer #1: Yes

Reviewer #2: Yes

2. Has the statistical analysis been performed appropriately and rigorously? 

Reviewer #1: Yes

Reviewer #2: Yes

3. Have the authors made all data underlying the findings in their manuscript fully available?

Reviewer #1: Yes

Reviewer #2: Yes

4. Is the manuscript presented in an intelligible fashion and written in standard English?

Reviewer #1: Yes

Reviewer #2: Yes

5. Review Comments to the Author

**Reviewer #1:**  • Although the abstract is well-structured and informative, it could benefit from improvement to enhance clarity and precision. Replacing vague terms like "major symptoms" and "multi-faceted approach" with more specific language would enhance the presentation of findings and recommendations. Additionally, refining "post-wartime" to "post-conflict" would align with more standard terminology. The methods section could also benefit from a brief mention of participant inclusion and exclusion criteria, as well as a clearer explanation of the systematic sampling process. These adjustments would strengthen the abstract's overall clarity and impact.

Introduction

• The introduction lays a solid foundation but could benefit from clearer focus and refinement. The detailed descriptions of GAD, PTSD, and depression (lines 47–58) could be shortened to emphasize their relevance in the context of wartime. Additionally, the discussion of the global burden of mental health disorders (lines 59–70) could more strongly highlight the importance of addressing these issues in developing countries like Ethiopia, particularly in the context of the North Ethiopian conflict. The section on infectious diseases (lines 75–76) seems unrelated to the study’s objectives and could be removed unless it directly supports the focus of the research.

• While lines 80–102 offer valuable insight into the impacts of war on women's mental health, there is some repetition, particularly regarding trauma and its effects. For instance, the mention of PTSD resulting from trauma (lines 56–57) and repeated references to stress caused by war (lines 90–93) could be combined to avoid redundancy. The introduction begins to focus on the North Ethiopian context only in lines 103–109. Shifting this earlier in the introduction would help establish the study's specific focus more quickly and create a stronger connection between the global and local perspectives.

Methods

• The methods section provides a thorough overview, but there are areas that can be improved for clarity and precision. In lines 115–123, while the setting is adequately described, there could be a clearer focus on how Woldia's context specifically impacts the study's objectives. The section on inclusion and exclusion criteria (lines 124–130) can be streamlined for brevity. For example, rather than listing each exclusion category separately, these could be grouped or summarized more succinctly. In lines 131–145, the sampling strategy is comprehensive, but the connection to the research aims could be made more explicit to ensure the sampling process is fully aligned with the study’s objectives.

• In the data analysis section (lines 201–219), the use of logistic regression and multivariable analysis is appropriate. However, lines 199–200 introduce the concept of effect-modifying variables, but the relationship between these variables and the study outcomes should be clarified further. This will enhance understanding of how these variables may influence the results. The ethical considerations (lines 220–225) mention informed consent, but elaborating on how participant confidentiality was maintained during data collection would strengthen this section. Additionally, tightening some sentences for conciseness and improving transitions between sections would improve the overall readability and flow.

Results

• In line 230, the sentence "A prominent segment, accounting for 41.7%, fell into the age group between 25 to 34 years, with a mean age of 36.2 + 10.2, spanning an age range of 18 to 60 years" could be broken into simpler sentences to improve readability. A clearer alternative could be: "The majority of participants (41.7%) were aged between 25 and 34 years. The mean age of participants was 36.2 years, with an age range of 18 to 60 years." This will improve sentence flow and make the information easier to digest. Similar improvements can be made in other areas where multiple statistics are stacked in one sentence, such as line 241: "The study revealed that over 37% of participants reported a monthly income below 1,500 Ethiopian Birr, with two-thirds earning less than 3,000 Ethiopian Birr (equivalent to $60)." This could be split for clarity: "Over 37% of participants reported a monthly income below 1,500 Ethiopian Birr. Two-thirds earned less than 3,000 Ethiopian Birr (approximately $60)."

• Ensure consistency in the formatting of percentages. In lines 236–237, for instance, the phrase "approximating ten percent" can be made more precise, e.g., "approximately 10%" or "10%". This ensures consistency in numerical representation across the section. Additionally, check for consistency in the presentation of data in tables and narratives (lines 235–236). Some data points, like "87.2%" and "10.7%" (line 234), could be followed by additional explanations to clarify their importance or relation to other figures.

• Transitioning smoothly from one section to another will enhance the coherence of the text. For example, in line 250, you move into economic challenges attributed to spouses’ alcohol and khat consumption. A brief sentence linking these findings to the previous socioeconomic section would help. For instance: "In addition to income and social status, other familial factors also played a significant role in the participants’ economic well-being, with some facing challenges due to spouses' substance use."

• The section on mental health (lines 256–273) could be more effectively organized by grouping similar findings together. For example, present all the mental health disorders in one subsection and break them into categories (e.g., depression, anxiety, PTSD) to improve clarity. Lines 257–266 could benefit from such organization: first, describe the general rates of depression, anxiety, and PTSD, then break them down further (e.g., percentage of participants with severe symptoms).

• The use of terms like "Oslo-social support index" (lines 248–249) should be briefly explained or defined, particularly for readers unfamiliar with these metrics. You could add a short explanation, such as: "The Oslo-social support index, which measures perceived social support, was used to assess the social well-being of participants."

Discussion

• The discussion provides important insights into the mental health challenges faced by women in Woldia after the conflict, but some areas can be refined to improve clarity and depth. First, the comparison with global averages (lines 340–344) is valuable, yet the argument could be strengthened by further contextualizing the impact of war-related stressors such as displacement, loss, and social disruption on mental health (lines 337–338, 345). Exploring how these stressors interact with individual characteristics like age, marital status, and socioeconomic factors (lines 333–335, 373–377) would offer a deeper understanding. Additionally, while the section on comorbidity (lines 359–363) is crucial, it would benefit from an expanded discussion on treatment strategies or interventions specifically designed for women with multiple mental health conditions, particularly in settings with limited resources (lines 365–366).

• The discussion of socio-demographic risk and protective factors (lines 371–380) aligns well with existing literature, but it would be helpful to delve into how these factors might present differently in conflict-affected areas (lines 373–376). For instance, exploring why certain protective factors, such as social support (lines 379–383), are more effective in post-conflict situations would provide additional insights. The suggestion of a multi-faceted intervention approach (lines 384–391) is commendable, but it would be even more valuable to describe how such strategies could be practically implemented in Woldia, offering a clear pathway for policymakers and practitioners. Lastly, while the study acknowledges its limitations (lines 392–399), it would be useful to suggest specific ways future research could address these gaps, such as conducting longitudinal studies to track changes in mental health over time (lines 395–396) and exploring the quality of social support networks (lines 397–398).

Conclusion and recommendation

• The conclusion and recommendations section effectively summarizes the findings and suggests practical steps to address the mental health crisis in Woldia. However, the conclusion could be enhanced by incorporating more specific details from the study's findings, such as the prevalence of comorbidity (lines 359–363) and its implications for treatment (lines 365–366), offering a more comprehensive understanding of the mental health challenges faced by women in the region. Furthermore, the recommendations could be more detailed by providing concrete actions or examples of successful interventions in similar contexts. This could include practical strategies for mobilizing multi-disciplinary teams and examples of effective community-based programs in other post-conflict areas. Such additions would strengthen the recommendations and provide actionable guidance for stakeholders addressing the mental health crisis in Woldia.

**Reviewer #2: ** Please see attached file.

6. PLOS authors have the option to publish the peer review history of their article (what does this mean? ). If published, this will include your full peer review and any attached files.

**Do you want your identity to be public for this peer review?** For information about this choice, including consent withdrawal, please see our Privacy Policy .

Reviewer #1: **Yes: ** Kindie Mitiku Kebede

Reviewer #2: **Yes: ** PD Dr. med. Andrea Jobst

---

## [Author Response · Author response to Decision Letter 1]

6 Mar 2025

Dear Editor and reviewers,

On behalf of our author group, we sincerely thank the Editor-in-Chief and the reviewers for their insightful and constructive feedback on our manuscript, “The Impact of Wartime Conflict on Mental Health Problems of Women in Conflict-Hit Populations in Woldia, Ethiopia.” The reviewers’ thoughtful comments and valuable suggestions have significantly enhanced the quality and clarity of our manuscript. We have carefully considered each point raised and have provided detailed responses, outlining how we addressed the concerns and tracking all revisions made accordingly. We truly appreciate the time and effort dedicated to reviewing our work.

In the attached file, we have structured our responses: the first column presents the reviewers’ comments, the second column contains our responses typically acknowledging and incorporating the suggested improvements, and the third column highlights the specific changes made in the manuscript, with modifications marked. We hope this format facilitates the review process. Once again, we sincerely thank you for your invaluable feedback and guidance.

Sincerely,

---

## [Editor Report · Decision Letter 1]

17 Mar 2025

The impact of wartime conflict on mental health problems of women in conflict-hit population in Woldia, Ethiopia

PONE-D-24-23457R1

Dear Dr. Deyessa,

We’re pleased to inform you that your manuscript has been judged scientifically suitable for publication and will be formally accepted for publication once it meets all outstanding technical requirements.

Kind regards,

Inga Schalinski

Academic Editor

PLOS ONE
---

## [Editor Report · Acceptance letter]

PONE-D-24-23457R1

PLOS ONE

Dear Dr. Deyessa,

I'm pleased to inform you that your manuscript has been deemed suitable for publication in PLOS ONE. Congratulations! Your manuscript is now being handed over to our production team.

Kind regards,

on behalf of

Dr. Inga Schalinski

Academic Editor

PLOS ONE